# Plant Antimicrobial Peptides (PAMPs): Features, Applications, Production, Expression, and Challenges

**DOI:** 10.3390/molecules27123703

**Published:** 2022-06-09

**Authors:** Olalekan Olanrewaju Bakare, Arun Gokul, Adewale Oluwaseun Fadaka, Ruomou Wu, Lee-Ann Niekerk, Adele Mariska Barker, Marshall Keyster, Ashwil Klein

**Affiliations:** 1Environmental Biotechnology Laboratory, Department of Biotechnology, University of the Western Cape, Bellville 7535, South Africa; 3255882@myuwc.ac.za (R.W.); 3056605@myuwc.ac.za (L.-A.N.); 4176676@myuwc.ac.za (A.M.B.); mkeyster@uwc.ac.za (M.K.); 2Department of Biochemistry, Faculty of Basic Medical Sciences, Olabisi Onabanjo University, Sagamu 121001, Ogun State, Nigeria; 3Department of Plant Sciences, Qwaqwa Campus, University of the Free State, Phuthadithjaba 9866, South Africa; Gokula@ufs.ac.za; 4Department of Science and Innovation/Mintek Nanotechnology Innovation Centre, Bio labels Node, Department of Biotechnology, Faculty of Natural Sciences, University of the Western Cape, Bellville 7535, South Africa; afadaka@uwc.ac.za; 5Plant Omics Laboratory, Department of Biotechnology, University of the Western Cape, Bellville 7535, South Africa

**Keywords:** PAMPs, biotechnology, structure, engineering, drug, modelling

## Abstract

The quest for an extraordinary array of defense strategies is imperative to reduce the challenges of microbial attacks on plants and animals. Plant antimicrobial peptides (PAMPs) are a subset of antimicrobial peptides (AMPs). PAMPs elicit defense against microbial attacks and prevent drug resistance of pathogens given their wide spectrum activity, excellent structural stability, and diverse mechanism of action. This review aimed to identify the applications, features, production, expression, and challenges of PAMPs using its structure–activity relationship. The discovery techniques used to identify these peptides were also explored to provide insight into their significance in genomics, transcriptomics, proteomics, and their expression against disease-causing pathogens. This review creates awareness for PAMPs as potential therapeutic agents in the medical and pharmaceutical fields, such as the sensitive treatment of bacterial and fungal diseases and others and their utilization in preserving crops using available transgenic methods in the agronomical field. PAMPs are also safe to handle and are easy to recycle with the use of proteases to convert them into more potent antimicrobial agents for sustainable development.

## 1. Introduction

Plants contain high concentrations of secondary metabolites, including tannins, quinines, terpenoids, phenols, and antimicrobial peptides produced by plant cells through metabolic pathways derived from primary metabolic pathways to generate an efficient defense system [1]. These secondary metabolites are produced in a response in the plant to ward off insects, bacteria, and fungi. Secondary metabolites have numerous effects (antibiotic, antifungal, and antiviral) against pathogens and possess UV-absorbing compounds to prevent serious leaf damage from light [2]. However, the knowledge that microbes have a reduced tendency to develop resistance toward antimicrobial peptides (AMPs) has made them more popular for use [3].

Antimicrobial peptides (AMPs) are cationic peptides capable of inhibiting protein transport, ion channels, or enzymes, acting as steroid hormone regulators, and interacting with DNA and RNA [4]. AMPs are widespread as host-defense peptides against pests and pathogens in many organisms [5,6]. They exist in diverse molecular forms but mainly as linear peptides from insects, animals, and plants [7]. However, polycyclic forms are produced by bacteria as lantibiotics. In contrast, other circular peptides are produced by other life forms such as bacteriocins (bacteria), cyclotides (plants), and theta-defensins (animals) [8,9]. Plant antimicrobial peptides (PAMPs) have evolved differently from other antimicrobial peptides (AMPs) due to the presence of cysteine residues, forming numerous disulphide bridges [10,11]. The disulphide bridges of cysteine-rich PAMPs can be cross-braced as cystine-rich peptides (CRP). PAMPs are a part of the plants’ barrier defense mechanisms isolated from roots, leaves, flowers, stems, and seeds of various species [5].

PAMPs have several characteristics (molecular weight, positive charge, and amphipathicity) similar to other peptides from insects, microbes, and animals. These characteristics are related to their defensive roles [12]. However, PAMPs have certain unique features, including a molecular weight between 2–6 kDa with two or six intramolecular disulphide bonds in the most PAMPs; family variation classified based on the presence or absence of cysteine motifs, sequence similarity, and conserved secondary and tertiary structure; compact structure; the derivation from ribosome with bioprocess precursors of three domains: N- and C-terminal pro-domains and a mature AMP domain [13].

The mechanism of PAMPs’ interaction with microbes is associated with cell lysis due to peptide penetration and disruption of lipid membranes and subsequent invasiveness of intracellular targets [14]. This involves producing and accumulating related peptides from storage organs and reproductive tissues of plants as the first line of defense against attack. PAMPs can also form ion channels to induce leakage of ions such as K^+^ and other intracellular contents, which cause inhibition of pathogen cell growth and cell death [15]. As a result, PAMPs have been termed promiscuous due to the different actions associated with the same structure. Several proposed mechanistic models are available, including the toroidal pore model, barrel stave model, and carpet model, to explain the different properties exhibited by PAMPs [16].

The biotechnological potentials of PAMPs have provided a novel source of drug discovery for the treatment of human infections and other diseases, from systemic therapy to topical administration [5]. Research into the chemical combination and modifications of targeted peptide residues would enhance PAMPs bioactivity, alongside biotechnology for discovering new peptides in host plants either as biocontrol agents or healthcare services. This review, therefore, aimed to explore the potential of PAMPs using the structure–activity relationship, discovery techniques, applications, and limitations to gain insight into their relevance in different fields. There are eco-friendly and efficient ways to improve the recycling for sustainable development and conversion of used PAMPs into high-valued antimicrobial products, such as the use of keratinolysis for keratin peptides recycling, in which keratinolysis hydrolysate was able to inhibit *E. coli* growth [17].

## 2. Structure-Activity Relationship of PAMPs

All AMPs are small-molecule polypeptides synthesized from ribosomes, in which their mature forms are cleaved from larger protein precursors with further post-translational modifications [18]. Some non-ribosomal synthetases can assist in the manufacturing of AMPs [19]. AMPs vary in structural forms between species but possess similar features such as shorter length, positive charges, and hydrophobic and hydrophilic regions [20]. Plant AMPs have diverse functions, structures, expression patterns, and specific targets, which provide more complex and difficult classifications [21]. PAMPs have a unique structure–activity relationship and are classified into different families based on sequence similarity, absence, or presence of cysteine motifs, with cysteine motifs determining their different disulfide bond patterns and tertiary structure folds [22]. The main families of plant AMPs include thionins, defensins, hevein-like peptides, knottin-type peptides, α-hairpinins, lipid transfer proteins, snakins, and cyclotides [23]. The ability of PAMPs to organize into specific families with conserved structural folds has allowed researchers to sequence variations of non-cysteine residues encased in the same scaffold to play different functions within a family [24]. PAMPs’ ability to tolerate hypervariable sequences while using a conserved framework allows them to recognize various targets by altering the sequence of non-cysteine residues [25]. The presence of disulfide bonds in most PAMPs augments their stability towards thermal, chemical, and enzymatic degradation, thus protecting their tertiary and quaternary structures [26].

In addition to having antimicrobial activity, PAMPs play a significant role in regulating plant growth and development and can be used as food additives [23]. PAMPs have a wide spectrum of antiviral, antifungal, antioxidative, antibacterial, chitinase, and proteinase-inhibitory activities [27] to create physical barriers against the spread and penetration of pathogens using waxy cuticle layers and trichomes, or using chemical barriers to inhibit the growth of the pathogens using a complex cell recognition system and photo hormone networks transcriptional pathways, secondary metabolites, and many diverse proteins [23]. PAMPs also act as prominent chemical barriers in plants to resist abiotic stresses [28]. The antimicrobial activity and cell selectivity of plant AMPs are influenced by many factors, including amino acid residues, net charge, hydrophobicity, amphipathicity, and structural propensity [29].

## 3. Discovery Techniques of PAMPs

Many tools are available to generate PAMPs using biological methods, with a subsequent activity prediction [30]. These peptides have been developed with reduced toxicity to human cells, with a great propensity for improved activity and stability to solve the continuous fight against food spoilage and waste [31]. PAMPs have been successfully extracted and purified from numerous plants, and their antimicrobial efficacy is characterized [32]. Peptides synthesized from this approach display improved activity with reduced cytotoxicity to human cells.

Apart from these, another method called biological synthesis involves using recombinant organisms to produce peptides [33]. The production of PAMPs through recombinant techniques is an efficient system to generate peptides, for example, plant defensin with a length of about 30–50 amino acid residues [34]. However, shortcomings of this method have been experienced, such as low production yields, inability to overcome the existing toxicity from the source, and failure to introduce non-biological components [35]. Fusion of such peptides with thioredoxin, chloroplast expression systems, and other cell vectors has been carried out to overcome such shortcomings, preventing cell lysis and assisting the formation of disulfide bonds [36]. Biological synthesis has successfully generated peptides such as potato AMPs, Snakin-1 (sn1), and Defensin-1 (pth1) [37].

Moreover, enzymatic hydrolysis is another technique available to generate PAMPs [38]. One or more enzymes are applied for the hydrolysis of a specific plant protein to generate short peptide sequences in the form of hydrolysates. Using this technique, another success has been recorded in producing antibacterial peptides from the fruit protein of *Fructus bruceae* through peptin hydrolysis [39]. Such peptides have been produced with antimicrobial properties through the hydrolyzing effect of alcalase and flavourzyme on the protein present in the seeds of *Salvia hispanica* [40]. Enzymatic hydrolysis is preferred to the use of microorganisms to carry out the hydrolyzing mechanism steps referred to as microbial fermentation due to ease of scalability and predictability within a short reaction time [41]. Microbial fermentation involves hydrolyzing a target protein through the action of an enzyme released by a microorganism to break down proteins into peptides for antimicrobial screening analysis [42].

PAMPs are important molecules which form part of the antimicrobial peptide databases, and are not described in detail as separate entities (Table 1). Some bacterial or fungal endophytes reside in plants where some peptides have been extracted. Abdelshafy Mohamad, Ma [43] isolated and characterized some beneficial endophytic bacterial populations associated with the medicinal plant Thymus vulgaris to alleviate salt stress and confer resistance in *Fusarium oxysporum*. In most plants, Bacillus subtilis, for instance, is a component of the endophytes of some plants which produce antimicrobial peptides, subtilisin, found in the antimicrobial peptide database, APD3 (https://aps.unmc.edu/AP/ (accessed on 20 February 2022)) with accession numbers AP00928. These peptides are being used for several in silico analyses, which is an active area of research and involves many computational tools either singly or combined to ease the discovery process and shorten the time of more putative peptides. de Azevedo dos Santos, Taveira [44] identified PAMPs with wide-spectrum antimicrobial and anti-inflammatory activities potent against diseases using computational technologies for their retrieval from the databases and afterwards described their characteristics. In addition, a few lines of research have used HMMER, a name given by the software developer Sean Eddy and Travis Wheeler, to identify motifs on proteome sequences for the discovery of novel peptides against HIV [45]. Tincho, Gabere [45] used a class of experimentally validated plant AMPs in his research and identified motifs on the proteome sequences to discover novel peptides against HIV using HMMER.

Overall, the choice of a peptide to be produced relies on many factors, including its toxic effects, haemolytic activities, and susceptibility to proteolysis. As such, the design and generation of PAMPs is a sensitive process that entails modification, cytotoxicity reduction, and improved safety to avoid detrimental effects on the entire antimicrobial characteristics and potency of the final product for assurance of stability.

## 4. Applications of PAMPs

### 4.1. Application of Plant Antimicrobial Peptides (PAMPs) in Biotechnology

The current antibiotics consumed or utilized have a higher possibility of producing resistant pathogens; thus, alternative approaches must overcome this drawback. One of such approaches is the implementation of PAMPs to alleviate pathogen infection and wound severity. The role of PAMPs is observed in various crops such as potatoes, tomatoes, and soybeans. The PAMPs are involved in the direct destruction of pathogen membranes and have an important role in the induction of genes in the Salicylic acid, Jasmonate-dependent pathways, and R-gene signaling [68]. A study by Niu, Zhong [69] employed PAMPs to strengthen *Glycine max* (soybean) production through effective control of Phytophthora root and stem rot (PRR) caused by the fungus *Phytophthora sojae*. In the past, fungicide application, improved soil drainage, and crop rotations were employed to control PRR in the field; however, these interventions are not sustainable. The use of *Capsicum annuum antimicrobial protein 1* (*CaAMP1*) to control *Xanthamonas campestris pv. Vesicatoria* in *Capsicum annuum* (pepper) using Agrobacterium-mediated transformation has proved beneficial. These overexpressed lines (*CaAMP1-OX*) were inoculated with *P. sojae race 1* and reported that *CaAMP1-OX* lines infected with *P. sojae* displayed higher tolerance to PRR, which was stable in the T4 generation when compared with the wild type. This study also reported that the plant disease resistance gene (R-gene), salicylic acid-dependent, and jasmonic acid-dependent genes were significantly upregulated after inoculation with *P. sojae*. Niu, Zhong [69] concluded that *CaAMP1-OX* plants drastically improved soybeans’ tolerance to PRR through prompt resistance responses controlled by multiple defense-signaling pathways.

Similar findings were reported by Herbel, Sieber-Frank [70], who focused on *Snakin-2* (*SN2*). *Snakin-2* is a PAMP, consistently expressed in plants’ aerial organs (leaves and flowers) (Figure 1). Like *CaAMP1*, *SN2* kills pathogens by targeting their phospholipid membranes. *SN2* forms pores in pathogens’ membranes and causes cell aggregation, halting the pathogen’s translocation within the plants [70,71].

Herbel, Sieber-Frank [70] investigated whether *SN2* expression responds to fungal infection by *Fusarium solani*, wounding, and exogenous application of phytohormones to heat, cold, and drought in *Solanum lycopersicum* (tomatoes) seedlings and mature plants. It was reported that fungal infection, wounding, and the application of phytohormones (specifically methyl jasmonate) strongly upregulated *SN2* gene expression after the stress was introduced, which was correlated to the mRNA levels of *SN2*. This work demonstrated that *SN2* likely plays a role in the methyl jasmonate signalling pathway. Furthermore, it was highlighted that the adjacent leaves of untreated matured tomato plants exhibited higher expression levels of *SN2* when compared with the treated leaf. The expression of *SN2* was recorded in other leaves, thus suggesting that long-distance signaling played a role in this defense system. A 25-fold increase in *SN2* expression was reported in the roots compared with its controls (after wounding), implicating that the plant produces antimicrobial substances in the entire plant to ensure complete protection against invading pathogens through a systemic defense response. This research further proved that the application of methyl jasmonate in tomato seedlings increased the protein level of bioactive *SN2* (Figure 2).

Transgenic *Brassica juncea* (oilseed) plants expressing the *msrA1* gene were screened for resistance against fungal pathogens such as *Alternaria brassicae* and *Sclerotinia sclerotiorum*. These pathogens are to affect the production of *B. juncea* crops negatively. Transgenic *B. juncea* plants infected with *A. brassicae* showed a 44%–62% reduction in hyphal growth, highlighting the pivotal role of *msrA1* in restricting fungi. A reduction in *A. brassicae* hyphal growth was also reported in the transgenic lines compared with the wild-type plants, representing 69%–85% disease protection in the transgenic lines. Similar results were observed for *S. sclerotiorum* infection in the transgenic lines, with approximately 56%–71.5% disease protection recorded compared with the wild type plants. In the wild type plants, the infection and spread of both phytopathogens were more rapid and prolific when compared with the transgenic lines of *B. juncea*. The transgenic lines restricted the spread of the phytopathogens, indicating the effectiveness of the activity of the *MsrA1* protein within these plants [73].

The biotechnological application of PAMPs establishes a promising approach to combating biotic and abiotic stresses that reduce crop yield. PAMPs also provide an environmentally and economically improved mechanism instead of the conventional approaches currently in use.

### 4.2. Application of Plant Antimicrobial Peptides (PAMPs) in Drug Design Using in Silico Modelling

Various algorithms exist to explore the protein–protein interactions of large contact surfaces through structure predictions and interaction with target protein complexes. In silico modeling has been used to generate more potent, cost-effective, and wide-spectrum PAMPs with computer-assisted design strategies to mitigate challenging problems. These include translating a primary sequence to peptide structures to solve various multi-drug-resistant pathogens [21]. One such strategy includes the generation of a 20-amino acid-bacteriocin peptide that can traverse the membrane of pathogens developed by Fields, Freed [74] using a machine learning approach and simple biophysical trait, causing cytotoxic, haemolytic, and antimicrobial effects. Brogi, Ramalho [75] explained computer-aided drug design (CADD) roles in identifying promising drug candidates such as PAMPs with cost-effectiveness and limited use of animal models in pharmacological research. This work repositioned marketed drugs and supported medicinal chemists and pharmacologists in rationalizing novel and safe drugs. Porto, Irazazabal [76] fortified guava with glycine using a genetic algorithm that yielded guavanin peptides and arginine-rich α-helical peptides. These peptides have bactericidal effects at low concentrations, which exert their efficacy through membrane disruption and hyperpolarization.

Apart from this, Li, Hu [23] emphasized using the different types of plant AMPs, the factors that affect their antimicrobial activities, mechanism of action, and potential application in the food industry, breeding industry, and agricultural and medical fields. An advanced computer-assisted design strategy that can prevent the complex challenges of relating primary sequence to these peptide structures with a concurrent delivery of more potent, cost-effective, broad-spectrum peptides as potential next-generation antibiotics was analyzed by Karwal, Vats [77]. They concluded that most AMPs have activity against microbes with modest direct antibiotic activity. Oyama, Olleik [78] identified two novel linear AMPs (hg2 and hg4) from a rumen metagenomics dataset with antibacterial activity against multi-drug-resistant *Staphylococcus aureus*. They used a classifier model design, a feature extraction method using molecular descriptors for amino acids to analyze, visualize, and interpret their activities. The peptide-bound methicillin-resistant *Staphylococcus aureus* (MRSA) in its lipids rather than human cell lipids indicates that hg4 may form a superior template as a therapeutic candidate for multi-drug resistant (MDR) bacterial infection.

In silico modeling is a crucial part of the drug design process, and its potential can enhance the drug design and optimization processes of PAMPs. In silico modeling can predict the structure of peptides lacking resolved 3-D structures, analyze the physicochemical properties to circumvent functional limitations, and screen peptide libraries to find new targets. These computational approaches can undergo continuous revision based on the evidence obtained from experimental studies. Molecular dynamics (MD) simulation is an in silico method for analyzing the physical movement of atoms and molecules, which are allowed to interact at a fixed period. This technique can reveal molecular interactions such as lipid–peptide interaction, the development of new antibiotic candidates, and the systemic comparison of activities with their mechanisms in pathogens [79,80].

### 4.3. Application of Plant Antimicrobial Peptides (PAMPs) in Medicine

The validation of PAMPs in clinical trials is faced with key challenges such as molecular mechanisms of PAMPs in various diseases, the significance of PAMPs in pharmaceutical industries, and the challenges in using PAMPs as therapeutics available in the market and PAMPs under clinical trials. [81]. Apart from the prospective therapeutic potential such as antiviral, antifungal, anti-mitogenic, anticancer, and anti-inflammatory properties, plant AMPs can act as immune modulators (Table 1). There is no doubt that PAMPs have the potential to replace conventional drugs to gain global drug market share. The efficacy of PAMPs due to their high specificity, low toxicity, and tight binding to their targets because of the large chemical space and the side chain variations of native amino acids accounts for the current databases having antimicrobial peptides of 7700, PAMPs inclusive [82]. The rapid spread of resistance genes in the environment is a cause for concern as it can lead to massive loss of life globally and the inevitable knock-on effects, which could lead to many countries’ economies becoming crippled [83]. Due to the sessile nature of plants, they are often attacked by microorganisms throughout their life cycle. This constant bombardment of biotic stress factors has led to the evolution of protective agents such as PAMPs, which effectively control bacterial and fungal infection and have gained the moniker “next generation of antibiotics” [68,84]. In a study by Oliveira, Souza [85], three synthetic AMPs (Mo-CBP_3_-PepI, Mo-CBP_3_-PepII, and Mo-CBP_3_-PepIII) were produced based on the PAMP Mo-CBP_3,_ which was the first isolated from *Moringa oleifera* seeds. The authors were able to show that the synthetic PAMP Mo-CBP_3_-PepIII exhibited no hemolytic or toxic activity to mammalian cells but still retained the ability to cause plasma membrane permeabilization in *Staphylococcus aureas*.

A study by Nganso, Sidjui [86] showed similar results when they extracted antimicrobial peptides from *Bauhinia rufescens* Lam, which form part of the *Fabaceae* family. The identified peptides were characterized by being cysteine-rich and showing antimicrobial activity. The antimicrobial peptides were effective against gram-negative bacteria and disrupted the cellular membrane [86]. The studies mentioned above clearly illustrate how plant-derived antimicrobial peptides can be used in the medical industry to control pathogenic organisms in humans and animals. The application of PAMPs for cancer treatment has been proposed by many researchers [7,87,88,89]. Yet, relatively few studies have been conducted to identify potential PAMPs and evaluate their therapeutic effects. Besides using PAMPs against cancer and microbial pathogens, researchers have also identified other peptides applications. Afroz, Akter (90) investigated the plant peptide (BRS-P19) isolated from *Bauhinia refescens*. The authors demonstrated that this therapeutic peptide produced inhibitory effects against the venom phospholipase A2 isolated from snake venom. The studies discussed in this section show that PAMPs have many applications within the medical sector, further reinforcing the need for more research into these beneficial plant-derived peptides.

### 4.4. Application of Plant Antimicrobial Peptides (PAMPs) in Agriculture

Due to the abuse or misuse of antibiotic compounds, antibiotic resistance is increasing in the agricultural industry. Research into using PAMPs has become an exciting new avenue [90]. Plants have long been exploited to express AMPs due to their ability to host recombinant proteins, relatively low cost, and high expression yields [91]. The AMPs expressed in plants were often identified in animal systems [92]; however, AMPs innate to plants have been observed in recent years (Table 2). A study by Inui Kishi, Stach-Machado [93] showed how PAMPs could be used to control bacterial disease in citrus. The study highlighted how a PAMP isolated from citrus (citrus-amp1) could effectively control *Xanthomonas citri* while presenting low toxicity, which is beneficial for any edible crops. During research to understand how these PAMPs perform their beneficial functions in plants, researchers have hypothesized that plant-signaling peptides may have evolved from ancient AMPs [94,95]. PAMPs may function by interacting with ROS (reactive oxygen species) and MAPK signaling to elicit a defense response against pathogen or pest attacks [94]. Another mechanism was shown in a study by Farkas, Maróti [96], who observed the ability of two PAMPs to cause direct membrane disruption in bacteria. Although this study used *Salmonella eterica* and *Listeria monocytogenes*, which are animal pathogens, for their tests, it shows the potential of PAMPs to affect bacterial phytopathogens directly. To identify potential PAMPs and elucidate the potential mechanisms, ongoing research is being conducted. A study by Ramada, Brand [97] investigated the bioactive peptides “encrypted” within known sequences of plant proteins. The antimicrobial peptides were evaluated for their antimicrobial activity, hemolytic activity, and their ability to inhibit basidiospores. The study showed that two antimicrobial peptides, Gr01 and Tc06, were able to display comparable control of *Ramularia areola* in cotton plants when compared to PrioriXtra, a known commercial fungicide (Table 2). These studies show the potential of PAMPs to be used to further improve bacterial and fungal resistance in commodity crops.

## 5. Scalable Production and Expression of PAMPs

### 5.1. Scalable Production of Plants AMPs (PAMPs)

Barashkova and Rogozhin [98] described the isolation and extraction of PAMPs under three major headlines: Plant material homogenization, extraction, and saturation and purification of the extract. The fractionation of the resulting extracts is carried out using liquid chromatographic techniques. The PAMPs can be sourced from different parts of the plant of choice, such as root, leaf, stem, bulb, flower, tuber, and whole plant and handled based on the physical characteristics of the part intended for use [99]. During extraction, the choice of the solvent or extractant is important. PAMPs extraction is carried out using water or water-based solutions (including salt, acid, and buffer) and organic-based solvents (such as ethanol) [100].

Proteases usually are utilized in hydrolysis reactions for the recycling of PAMPs to convert them into more potent antimicrobial agents by creating specific conditions (for example, excess of the substrate to the enzyme in solution), and the inverse reaction can occur, which causes dehydration synthesis of amide bonds that ultimately leads to the formation of peptides (aminolysis). This system can be controlled through either thermodynamics or kinetically [101]. In a thermodynamic-controlled system (TCS), the presence of protease does not change the equilibrium of the reaction but rather acts as a catalyst and improves the overall reaction rate. The protease will be used entirely, and the free carboxyl group will donate an acyl group forming the acyl intermediate (Figure 3). All types of proteases are compatible in this system; the problem, however, with synthesizing peptides using this system, is that the formation of the acyl intermediate from the carboxyl group is very slow and often results in very low product yield OH: product of hydrolysis [102].

### 5.2. Expression of PAMPs

The expression of peptide/amino acid residues is an important phenomenon during synthesis, which starts with the selection of a host strain. *E. coli* BL21 (DE3) is commonly used for recombinant peptide synthesis due to the absence of proteases ompT and Ion, which leads to peptide deterioration. However, the selection of a host strain ultimately depends on the structural properties of the peptide. For example, *E. coli* BL21 struggles to synthesize peptides that contain high amounts of disulfide bonds, forming protein aggregates instead. Using other *E. coli* strains such as Origami or Rosetta-gami, which contain negative mutations in thioredoxin reductase (trxB) and glutathione reductase (gor) genes, will facilitate the expression of peptides which are rich in disulfide bonds [103]. This system’s main concern is that unnatural amino acids cannot be produced using this approach. There are ways to circumvent this by linking the PAMP with other proteins like thioredoxin, which results in other challenges such as low production yield [104].

PAMPs expressed this way need to be linked with a fusion tag creating a fusion protein, which is important to prevent peptide toxicity to the host and proteolytic degradation, leading to a low yield of PAMPs (Figure 4). PAMPs in the form of the fusion protein can be isolated by cleaving off the fusion tag at the carrier-peptide junction [105]. In addition, more than one fusion tag (i.e., spacer, cleavage site, and affinity tag) should be used to increase the efficiency of downstream processes such as purification and separation of PAMP from fusion tag. Small ubiquitin-related modifier Smt3 (SUMO) is a popular fusion tag used in the recombinant peptide synthesis method. This fusion tag is commonly used with the hexahistidine (His6) affinity tag. The His-tag will come into play during chromatography-based purification; in addition, SUMO protein has a hydrophobic core structure, which improves the solubility of synthesized PAMP [106]. Another advantage is that SUMO fusion protein does not require the addition of a cleavage site as SUMO fusion protein is recognized by SUMO protease (Figure 4).

## 6. PAMPs Genome Transcriptome and Proteome

PAMPs production and prediction are possible using RNA sequencing and de novo transcriptome assembly to initiate studies based on AMP gene evolution and expression. This allows the detection of strong peptide candidates to be used in drug discovery and other biotechnology products using transcriptomics and proteomics technologies. Rodriguez-Decuadro, da Rosa [108] identified 78 AMPs transcripts from *Peltophorum dubium*, using whole transcriptome sequencing coupled with de novo transcriptome assembly. These AMPs were classified into five families, including hevein-like, lipid-transfer proteins (LTPs), alpha hairpinins, defensins, and snakin/GASA (Giberellic Acid Stimulated in Arabidopsis) peptides. Noonan, Williams [109] investigated the AMP genes associated with fungus and insect resistance in maize to facilitate the breeding of host plant resistance and improve maize production, where 39 new maize AMPs were identified in addition to the seven known maize AMPs. mRNA expression analysis of the maize AMP genes was carried out using quantitative real-time polymerase chain reaction (qRT-PCR), where five of the maize AMP genes were associated with insect or fungus resistance. Yakovlev, Lysøe [110] carried out transcriptome profiling and in silico detection of AMPs of red king crab, *Paralithodes camtschaticus* to contribute to the use of AMP in the development of drug candidacy to alleviate antibiotic resistance, which is a global health threat. This work generated a transcriptome data set and AMPs to provide a solid baseline for further functional analysis and contribute future application of red king crab as a bio-source and its use as a seafood delicacy.

Umadevi, Soumya [111] described the AMP signature profile of black pepper and their expression upon Phytophthora infection using a label-free quantitative proteomics strategy where differential expression of 24 AMPs revealed a success of a combinatorial strategy in the defense network. The work offers great promise for the use of plant AMPs in the regulatory processes of evolutionary importance to exploit them as next-generation molecules against pathogens. Ngashangva, Mukherjee [112] analyzed the AMP’s metabolome of bacterial endophyte isolated from the traditionally used medicinal plant *Millettia pachycarpa* Benth. The outcome of this work revealed that both genomic and proteomic results would substantially increase the understanding of AMPs and assist the discovery of novel biological agents. Thus the potential applicability of AMPs in transcriptomics and proteomics revealed additional roles of these molecules in the regulation of plant growth, drug development, and treatment of diseases under high efficacy apart from their antimicrobial roles.

## 7. Challenges of PAMPs

Several persistent problems remain with the effective utilization of PAMPs to bind and identify their specific target. Tang, Prodhan [113] attributed one of these to the complexity associated with the purification of specific PAMPs due to the diversity of screening, identification, and purification methods. The problem is exacerbated by the fact that AMPs can be produced in almost all plant organs. Additionally, in a study by Rogozhin, Ryazantsev [53], wild type grains produced more AMPs than cultivated types due to higher variability of C-terminal fragment sequences and a higher percentage of hydrophobic amino acids in the wild grain AMPs than the cultivated ones, making it difficult to produce active plant AMPs in large quantities. Apart from this, because plants are cultivated differently, the expression of an AMP from a plant can be difficult due to artificial cultivation using transgenic technology on a commercial scale. However, endogenous degradation of AMPs by proteases in the leaf cells can be reduced by single amino acid substitution of AMP modification. However, the proteases that reside in the leaf intracellular fluid can be used to achieve expected transgenic functions [114]. The selectivity of PAMPs is another challenge limiting their activities for binding or killing target cells rapidly by perturbing the integrity of the plasma membrane. However, the structure of peptides and their activity relationship needs to be further understood to solve this problem in order to design more novel PAMPs from the existing ones [7].

## 8. Conclusions

PAMPs have proven to be vital biotechnology tools due to the safety and ease of recycling, such that enormous investments are being allocated to genetic engineering for their use in the production of insecticides and the improvement of transgenic products already resistant to pathogens. PAMPs exhibit several compensatory advantages compared with current antibiotic drugs because they possess a naturally occurring defense mechanism used by plants since antiquity in fighting pathogenic challenges. PAMPs also have good applicability in the quest for new medicines for human health, eliminating pests’ spread, and improving agribusiness food production. To this end, these underutilized PAMPs and their products offer great promise as a novel source of drug discovery for treating human infections and other diseases to solve myriad problems confounding pathogen resistance and lack of sensitivity of antibiotics.

## Figures and Tables

**Figure 1 molecules-27-03703-f001:**
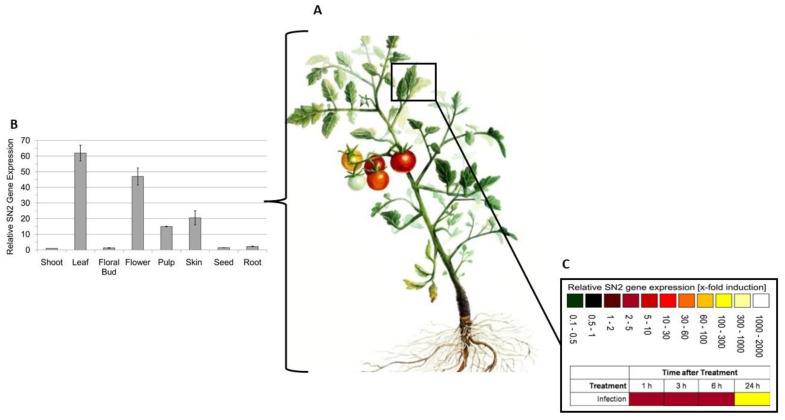
Relative gene expression of Snakin-2 in Solanum lycopersicum plants. (**A**) Diagram of a Solanum lycopersicum plant (Tomato) at the fruit-bearing stage. (**B**) Relative gene expression levels of SN2, using qRT-PCR, in healthy adult S. lycopersicum plant. SN2 recorded the highest expression in the leaves and flowers, respectively; therefore, further analysis of SN2 activity was conducted on the leaves [70]. (**C**) Relative gene expression levels of SN2, using qRT-PCR, after infection of tomato plant with pathogenic F. solani. The heatmap represents an upregulation of SN2 expression over 24 h, with the expression of SN2 being highest at the 24 h. This illustrates that the presence of F. solani in the tomato plants strongly increases the expression of the SN2 gene as a defense against pathogenic attack [70].

**Figure 2 molecules-27-03703-f002:**
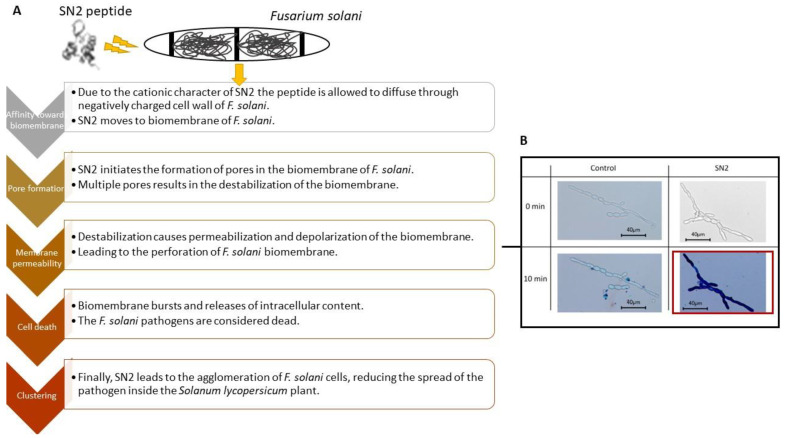
Schematic diagram of the mode of action of SN2 peptide on *Fusarium solani* pathogen. (**A**) The cationic SN2 peptide has a high affinity for the negatively charged cell wall of *F. solani*, diffusing through the cell wall to initiate pore formation on the biomembrane. This ultimately results in the rupture of the biomembrane, releasing the intracellular contents of the pathogenic cell, killing the *F. solani* pathogen [71,72]. (**B**) Cell viability assay conducted on *F. solani* pathogen treated with SN2 peptide. Trypan Blue dye (0.5%) was used to test the cell viability by infiltrating pathogenic cells with damaged biomembranes and staining these cells. This microscopic image represents that after 10 min of exposing F solani to the SN2 peptide, most of the SN2-treated cells were stained blue compared with the control, indicating SN2 disrupted the biomembrane allowing the dye to penetrate the SN2-treated cells, killing the pathogenic cells [71].

**Figure 3 molecules-27-03703-f003:**
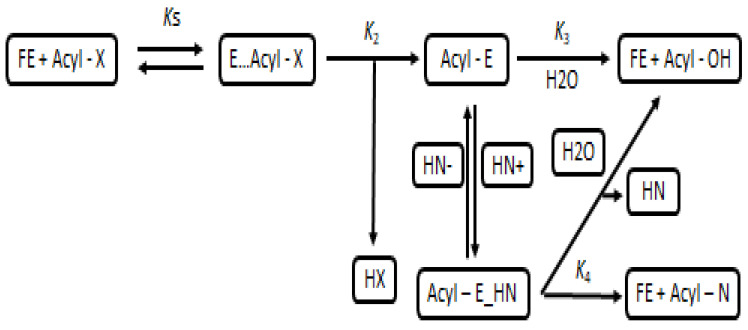
Kinetically control enzymatic peptide synthesis. FE: free enzyme, Acyl-X: acyl donor substrate, E., Acyl-X: acyl-enzyme complex, HX: released group, Acyl-E: acyl-enzyme intermediate, HN: nucleophiles, Acyl-N: target peptide, Acyl-OH: product of hydrolysis.

**Figure 4 molecules-27-03703-f004:**
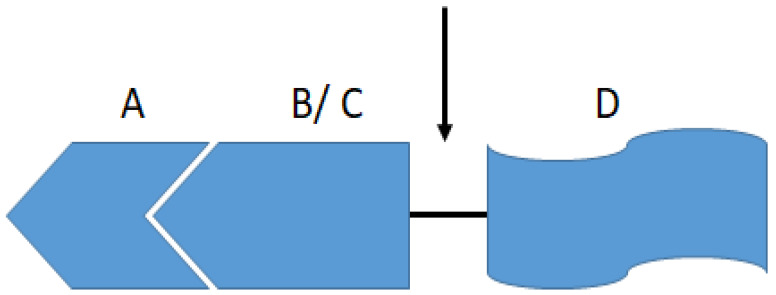
SUMO fusion protein. **A**: fusion tag 1, **B**: fusion tag 2, **C**: cleavage site, **D**: PAMP. For example, four interacting SUMO motif domains on the SUMO-ubiquitin E3 ligase RNF4 identify more than 300 peptides in HeLa cells using heat-shock treatment [107].

**Table 1 molecules-27-03703-t001:** Description of novel plant antimicrobial peptides (PAMPs), their sources, and respective function.

S/N	PAMPs	Source	S-S Bonds	Number of Amino Acid Residues	Function	References
**1**	Snakin-1	*Solanum tuberosum*	6	88	Antimicrobial activity with rapid aggregation of both Gram-positive and Gram-negative bacteria	[46]
**1**	Snakin-2	*Solanum tuberosum*	6	104	contribute to the biochemical stability in response to biotic (i.e., induced by bacteria, fungi, and nematode pathogens) and abiotic (salinity, drought, and ROS) stressors, as well as in crosstalk promoted by plant hormones, with emphasis on abscisic and salicylic acid (ABA and SA, respectively)	[47]
**2**	Trypsin inhibitor-2	*Momordica cochichinensis*	3	29	improve bioactivities by increasing stability and lowering flexibility as well as sensitivity to proteolytic attack	[48]
**3**	Trypsin inhibitor-1	*Momordica cochichinensis*	3	38	improve bioactivities by increasing stability and lowering flexibility as well as sensitivity to proteolytic attack	[49]
**4**	Antimicrobial peptide 1/AMP1_COCNU (P86705)	*Cocos nucifera*	Nil	9	Antibacterial	[50]
**6**	Antimicrobial peptide 2/AMP 2 (P86706)	*Cocos nucifera*	Nil	11	Antibacterial	[51]
**7**	Antimicrobial peptide 3/AMP 3 (P86707)	*Cocos nucifera*	Nil	8	Antibacterial	[50]
**8**	*Cycas revoluta*–anticancer peptide 1/Cr-ACP1	*Cycas revoluta*	Nil	9	inhibits cell proliferation and induces apoptosis in cancer-derived cell lines	[52]
**9**	*Pharbitis nil*/Pn-AMP1 (P81591)	*Ipomoea nil*	5	41	Chitin-binding protein with a defensive function against numerous chitin-containing fungal pathogens. It is also an inhibitor of Gram-positive bacteria such as *B. subtilis*	[53]
**10**	*Pharbitis nil* /Pn-AMP2 (P81591)	*Ipomoea nil*	5	41	Antibiotic, Antimicrobial, Fungicide	[54]
**11**	*Fagopyrum esculentum*-antimicrobial peptide 1/Fa-AMP1 (P0DKH7)	*Fagopyrum esculentum*	5	40	active against plant pathogenic fungi and Gram-negative and -positive bacteria	[55]
**12**	*Fagopyrum esculentum*-antimicrobial peptide 2/Fa-AMP2 (P0DKH8)	*Fagopyrum esculentum*	5	40	Antifungal and antibacterial (both Gram+ and -)	[55]
**13**	*Amaranthus caudatus*–antimicrobial peptide 1/Ac-AMP1 (Q9S8Z6)	*Amaranthus caudatus*	3	29	Chitin-binding	[56]
**14**	*Amaranthus retroflexus*–antimicrobial peptideAr-AMP (Q5I2B2)	*Amaranthus retroflexus*	3	89	inhibits the growth and induces morphological changes in fungal pathogens	[57]
**15**	Antimicrobial peptide 1.1a/AMP-1.1a (E1UYT9)	*Stellaria media*	7	167	Antifungal	[58]
**16**	Antimicrobial peptide 1.2a/AMP-1.2a (E1UYT9)	*Stellaria media*	7	167	Antifungal	[58]
**17**	*Stellaria media* antimicrobial peptide 3/SmAMP3 (C0HJU5)	*Stellaria media*	3	35	Antifungal	[58]
**21**	Cowpea-thionin/Cp-thionin II	*Vigna unguiculata*	4	46	Antifungal and antibacterial activity	[59]
**27**	Thionin 2.4	*Arabidopsis thaliana*	3	134	Antifungal activity	[60]
**28**	*Solanum tuberosum*-Snakin 1/StSN1	*Solanum tuberosum*	6	88	Antifungal, antiyeast and antibacterial activity	[61]
**31**	Cycloviolacin O2	*Viola odorata*	3	30	Antifungal and antibacterial activity	[62]
**32**	Cycloviolacin O8	*Viola odorata*	3	118	Antifungal and antibacterial activity	[62]
**33**	*Phytolacca Americana*/PAFP-S	*Phytolacca americana*	3	65	Antifungal activity	[54]
**34**	*Mirabilis jalapa*-antimicrobial peptide 1/Mj-AMP1	*Mirabilis jalapa*	3	61	Antifungal activity	[63]
**35**	*Mirabilis jalapa*-antimicrobial peptide 2/Mj-AMP2	*Mirabilis jalapa*	3	63	Antifungal activity	[63]
**37**	*Eucommia ulmoides* antifungal peptide 1/EAFP1	*Eucommia ulmoides*	5	41	Antifungal activity	[64]
**38**	*Eucommia ulmoides* antifungal peptide 2/EAFP2	*Eucommia ulmoides*	5	41	Antifungal activity	[64]
**39**	*Helianthus annuus*-antimicrobial peptide 10/Ha-AP10	*Helianthus annuus*	4	116	Antifungal activity	[65]
**40**	*Capsicum annuum* lipid transfer protein 1/CaLTP1	*Capsicum annuum*	Nil	114	Antifungal activity	[66]
**41**	Mung bean nsLTP	*Phaseolus mungo*	4	91	Antifungal and antibacterial activity	[67]

**Table 2 molecules-27-03703-t002:** Applications of PAMPs in agriculture.

S/N	PAMPs	Application	References
1	Snakin-2	Biochemical stability in response to biotic and abiotic stresses	[47]
2	Antimicrobial peptide 1/AMP1_COCNU (P86705)	Antibacterial activities	[50]
3	Antimicrobial peptide 2/AMP 2 (P86706)	Antibacterial activities	[51]
4	Antimicrobial peptide 3/AMP 3 (P86707)	Antibacterial activities	[50]
5	*Pharbitis nil*/Pn-AMP1 (P81591)	Defensive function against numerous chitin-containing fungal pathogens. It is also an inhibitor of Gram-positive bacteria such as *B. subtilis*	[53]
6	*Pharbitis nil*/Pn-AMP2 (P81591)	Antimicrobial and fungicidal activities	[54]
7	*Fagopyrum esculentum*-antimicrobial peptide 1/Fa-AMP1 (P0DKH7)	Defensive function against plant pathogenic fungi and Gram-negative and -positive bacteria	[55]
8	Cowpea-thionin/Cp-thionin II	Antifungal and antibacterial activity	[59]
9	Thionin 2.4	Antifungal activity	[60]
10	*Solanum tuberosum*-Snakin 1/StSN1	Antifungal, antiyeast, and antibacterial activity	[61]
11	Cycloviolacin O2	Antifungal and antibacterial activity	[62]

## Data Availability

Not applicable.

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
