# Peer review of "Plant Antimicrobial Peptides (PAMPs): Features, Applications, Production, Expression, and Challenges"

_molecules, 2022, doi:10.3390/molecules27123703_

Round 1

Reviewer 1 Report

The work is devoted to the use of new plant antimicrobial peptides (plant-AMPs) in biotechnology. The topic is interesting for a wide range of readers. Unfraternally, this manuscript has fundamental flaws.

In general, this review has little relevance to plant-AMPs, since the authors did not conduct a thorough search and analysis of scientific articles relevant to the stated topic. At the same time, noticeably numerous and unreasonable self-citations.

Almost all the time, the focus of the authors shifts from plant peptides to antimicrobial peptides in general (bacteriocins, AMPs of animal origin, etc.). I will list here all the places where it is noticeable.

  • Lines 74-76. Plant peptides have unique structure, which leads to the features of their mechanism of antimicrobial action, which differs from bacteriocins, animal defensins, etc. Authors should provide a detailed description of the relationship between the structure of plant-AMP and the mechanism of its antimicrobial action. This mechanism will be different in the case of a bacterium, a fungus or a virus. The authors should not cite their own work describing the hypothetical mechanism of pore formation (which, by the way, is known and described many years ago).
  • Lines 124-125. Here you could describe the extraction methods in more detail (see https://doi.org/10.1186/s13007-020-00687-1).
  • Lines 127-129. What is the relationship exist between plant-AMPs and human milk protein, lactoferrin etc ? If there are no examples for plant-AMPs, then you should not write about proteolysis as a source of novel AMP at all. Please stick to the stated topic of your Review.
  • Lines 131-137. Chemical synthesis is not related with discovery of plant-AMPs. It is a method of AMP obtaining. If you do not agree, then give some examples of the chemical synthesis of structures resembling plant antimicrobial peptides.
  • Lines 166-180. In silico prediction of novel AMPs is a modern and perspective approach, however, this review does not contain any cited works related to plant-AMPs. What does viral protein sgp have to do with plant-AMPs? The only one reference to de Azevedo dos Santos et al. (2020) is given, but cited work is not relevant, I checked it and there is no any word about in silico discovery.
  • Table1. If the table lists novel plant-AMPs, then the works cited should be in the range of 2021-2017. The works of 2008 are too old.
  • Figure 2 (listed as Figure 1). How  “Schematic diagram of the mode…” consistent with text of this paragraph (specifically Lines 246-248)? Check this.
  • Section 4.3 described investigations of plant-AMPs action against various pathogens and cancel cells in laboratory conditions rather than its real medical application. It makes sense here to discuss the reasons why clinical trials of plant-AMPs are difficult. If such reasons exist.
  • The section 5 could be deleted, which will not affect the structure or meaningful of this Review, since this section describes the general principles of peptide synthesis without focusing on plant peptides.
  • The section 5.3 -- I did not found how content of this section related to the purification of plant peptides? This Section could be deleted, since general principles of RP-HPLC, Ion exchange chromatography and other purification techniques are well described in appropriated textbooks and reviews, articles, and protocols.

Author Response

Thank you for your feedback. We are grateful. We hereby attach feedback to your comments as attached.

Reviewer 2 Report

Dear Editor,

The manuscript entitled “Applications and Relevance of Novel Plant Antimicrobial Peptides (PAMPs) in Biotechnology: An Overview” proposes to review the highly interesting topic of plant antimicrobial peptides and its biotechnological exploitation. However, after carefully reading the manuscript, I found that there are several major issues with this review that point me to recommend not publishing it in this state.

Major issues:

  • I do not find any interconnection between the title, the abstract and what is discussed in the text. While it is proposed that the biotechnology exploitation will be explored, just one or two paragraphs are dedicated to the theme. Most of the text and images are dedicated to techniques of chemical synthesis.
  • As discussed above, part of the text is dedicated to chemical synthesis, and clearly stating that is not feasible at industrial scales for exploitation in agriculture. The text should be balanced accordingly to the topic that you want to demonstrate
  • The use of the topic “cancer” in the abstract is misleading in several ways: first is almost not discussed in the text; then it is not the first application you think of with peptides with antimicrobial activity; and finally, it should be pointed out that the utilization of these peptides for cancer treatment have several issues associated.
  • I find hard to understand the choice of figures and tables in this manuscript. The figures seems to be a collection of data from one study and a tutorial on chemical synthesis. The tables show more effort from the authors in bibliographic research, but I do not understand the choice of certain peptides and references in detrimental of hundreds of others. Even table 1 claims that are “novel” peptides described, but with references from 2008 and 2014.

Minor issues:

  • Text should be revised for English in several sentences
  • Several typos were found. Examples:
    • Figure 1 repeated two times
    • Page 1 38 – to to
    • Page 1 36 ‘metabolic pathways’ two times in a row.

Best regards,

Author Response

Thank you for your feedback; we really appreciate it. We hereby provide corrections to your comments as attached.

Reviewer 3 Report

The submitted manuscript summarised the recent development of plant antimicrobial peptides. There are a few comments that need to be addressed.

  1. Some of the font sizes of the figures are too small, they should be revisited.

2,           line 51, a very good ref on cyclotide should be cited. (Curr Protein Pept Sci . 2004 Oct;5(5):297-315.doi: 10.2174/1389203043379512.)

3,           section 4.2, the in silico modelling discussion. Molecular dynamics simulations are a powerful tool for antimicrobial study, and they should be discussed here.

For example, Biochim Biophys Acta . 2012 May;1818(5):1402-9.doi: 10.1016/j.bbamem.2012.02.017.

Infect Immun . 2021 Mar 17;89(4):e00703-20. doi: 10.1128/IAI.00703-20.

Eur J Med Chem . 2022 Mar 5;231:114135. doi: 10.1016/j.ejmech.2022.114135.

4,           The references style might need to be revised.

Author Response

Thank you very much for your feedback; we are highly grateful. We hereby attach corrections to your comments as attached.

Reviewer 4 Report

Title: Applications and Relevance of Novel Plant Antimicrobial Pep- 2 tides (PAMPs) in Biotechnology: An Overview

The review entitle “Applications and Relevance of Novel Plant Antimicrobial Pep- 2 tides (PAMPs) in Biotechnology: An Overview” summarized relevance and applications of PAMPs using its structure-activity relationship. Moreover, describes the correlation of discovery techniques aspect to identify these peptides and present them as a complementarity. The Reviewer find this approach relevant and actual since the environmental problem increase daily. However, few aspects I suggest to take into account.

Please provide to the introduction section slight discussion regarding the recycling process and sustainable development to make the present research more relevant in context of environmental problem. Please also discuss this finding shortly in the discussion section in context of why such peptides has crucial impact in this direction. Therefore, after, the abstract and conclusion sections should be also correlated.

Additionally, please improve the resolution of all figures.       

Author Response

(The authors gave the same response as above.)

Round 2

Reviewer 1 Report

The manuscript has become better, but there is still a need to check for typos.

Author Response

Many thanks for your feedback. We really appreciate the quality of your review.

Reviewer 3 Report

The authors have addressed most of my comments. But the Font size in figure 1 and 2 are still hard to read.

Author Response

Many thanks for your feedback. We really do appreciate the quality of the review.
